# Thermal and Calorimetric Investigations of Some Phosphorus-Modified Chain Growth Polymers 1: Polymethyl Methacrylate

**DOI:** 10.3390/polym14071447

**Published:** 2022-04-01

**Authors:** Malavika Arun, Stephen Bigger, Maurice Guerrieri, Paul Joseph, Svetlana Tretsiakova-McNally

**Affiliations:** 1Institute of Sustainable Industries and Liveable Cities, Victoria University, P.O. Box 14428, Melbourne, VIC 8001, Australia; stephen.bigger@vu.edu.au (S.B.); maurice.guerrieri@vu.edu.au (M.G.); paul.joseph@vu.edu.au (P.J.); 2Belfast School of Architecture and the Built Environment, Ulster University, Newtownabbey BT37 0QB, UK; s.tretsiakova-mcnally@ulster.ac.uk

**Keywords:** polymethyl methacrylate, phosphorus-containing groups, additive and reactive routes, thermal degradation, calorimetric evaluations

## Abstract

The thermal and calorimetric characterizations of polymethyl methacrylate-based polymers are reported in this paper. The modifying groups incorporated the phosphorus atom in various chemical environments, including oxidation states of III, or V. Both additive and reactive strategies were employed, where the loading of phosphorus was kept at 2 wt% in all cases. The plaques, obtained through the bulk polymerization route, were subjected to a variety of spectroscopic, thermal and combustion techniques. The results showed that the different modifying groups exerted varying nature, degrees and modes of combustion behaviors, which also included in some cases an additive, and even an antagonistic effect. In the case of covalently-bound phosphonate groups, early cracking of the pendent ester moieties was shown to produce phosphoric acid species, which in turn can act in the condensed phase. For the additives, such as phosphine and phosphine oxide, limited vapor-phase inhibition can be assumed to be operative.

## 1. Introduction

Polymethyl methacrylate (PMMA) is an important member of the class of acrylic polymers, which is also used widely for industrial and household purposes. It has excellent properties such as good transparency, high impact strength, relatively high resistance to chemical attack and weathering, and UV resistance [1,2]. Furthermore, copolymerization reactions of PMMA with other monomers can yield hybrid materials that often exhibit some additional and advantageous properties. In spite of the several desirable attributes of PMMA, the relatively high flammability is a serious problem that often limits its wider applicability as a versatile thermoplastic material. When subjected to an external heat flux, methacrylic polymers generally undergo extensive thermal degradation. This primarily occurs through depolymerisation of PMMA via the unzipping of the polymer chain, resulting in almost complete production of its monomer, methyl methacrylate [3,4,5].

Generally, at higher temperatures, a series of complex reactions also can take place, resulting in the formation of decomposition products such as butene, methacrylic acid anhydrides, etc. [6].

Given the inherent fire hazard associated with synthetic polymeric materials, there were several attempts to reduce the ignitability and fire growth of its finished products- here both the additive and reactive strategies were employed with varying degrees of success with different classes of polymers [7]. Generally, phosphorus-based fire retardants (FRs) and their combustion products are less toxic than the corresponding halogen-based ones [8,9]. Furthermore, a large number of these compounds are available with a difference in their chemical environments of the phosphorus atom [10]. The efficiency of phosphorus-containing compounds in the additive, as well as the reactive categories, is also well documented [10,11].

The efficiency of phosphorus-based compounds generally depends on several factors, such as the chemical environment and oxidation state of the P atom; volatility; nature of the decomposition products formed upon thermolysis; etc. [12,13,14]. These compounds when subjected to degradation are shown to produce substituted phosphoric acid species that could inhibit the combustion process in the condensed phase. The condensed-phase activity of phosphorus compounds, predominantly, involves char formation which in turn is facilitated by the dehydration of the polymeric structure leading to cyclization, cross-linking and aromatization/graphitization [10]. Cross-linking can be also induced by the decomposition by-products of the phosphorus compounds. Phosphorus compounds, depending on their chemical nature, can also exert a noticeable degree of vapour activity. Various types of chemical moieties (mainly oxygenated species), emanating from substituted phosphoric acid species in the vapour phase are also identified [15]. 

Furthermore, halogenated and phosphorus-containing compounds when combined together are shown to exhibit some degree of cooperative interaction similar to the halogen/antimony combination [7]. Phosphorus- and nitrogen-containing compounds, when combined, can also exhibit such an effect [16]. However, this effect can also depend on the chemical nature of the nitrogen atom in the modifying group (for example, amide, imide, nitrile, etc.) when present with a particular type of phosphorus-bearing moiety (for instance, with phosphonate esters) [17]. In addition, the extent of the synergetic effect can also depend on whether both the atoms form part of the same pendant group [18]. In addition, P/N- containing compounds such as phosphoryl amino esters and phosphoramides are also tested for enhanced flame retardant properties [19].

Given the renewed interest in phosphorus-containing FRs, especially as alternatives to environmentally harmful halogenated compounds, the desirable attributes of several classes of them were explored through the current work. Thus, the additives used in the present study, included: phosphine, phosphine oxide, 9,10-dihydro-9-oxa-10-phosphaphenanthrene 10-oxide (DOPO), phosphites, phosphate and phosphonates. In the reactive category, polymerizable compounds, such as aliphatic and aromatic phosphonates, phosphorus/nitrogen- (P/N-) containing and phosphate esters were employed. The above compounds incorporate the phosphorus atom in different chemical environments and oxidation states (i.e., oxidation number III, or V) (Table 1). The novelty of the present study stems from the fact that it has also attempted to identify the variation in the flame retardation, if any, brought about by the differences in the chemical nature and/or oxidation state of the phosphorus atom, by essentially keeping the same loading (2 wt% with respect to phosphorus), amongst a variety of additives/reactives. 

In the following table (Table 2), the respective masses of the additives/reactives for MMA are given.

## 2. Materials and Methods

### 2.1. Materials

All the chemicals, reagents and solvents used in the present study were purchased from Aldrich Chemical Company, except the following: 9,10-dihydro-9-oxa-10-phosphaphenanthrene 10-oxide (DOPO) and diethyl-1-propylphosphonate (Thermofisher Scientific, Melbourne, Australia). Generally, the solid compounds were used as received, whereas liquid reagents and solvents were, optionally, dried by keeping them over molecular sieves (4 Å). Furthermore, thermally labile initiators and monomers were stored under sub-ambient temperatures in a refrigerator, or a freezer, as the case may be. The inhibitors (typically hindered phenolic compounds, such as hydroquinone monomethyl ether), were removed from methyl methacrylate by passing through proprietary inhibitor removal columns, purchased from Aldrich Chemical Company, Melbourne, Australia. The precursors and the comonomers used in the present study were synthesized by using previously reported procedures [20,21,22,23], which also included the additive, diethylbenzylphosphonate.

### 2.2. Synthesis of the Additive (Diethylbenzylphosphonate)

The additive was synthesized using the *Michaelis-Arbuzov* reaction. The required amount of benzyl bromide (23.8 mL, 0.2 mol) was mixed with triethylphosphite (34.3 mL, 0.2 mol), and the mixture was refluxed at 90 °C for 8 h, followed by heating at 140 °C for an extra 2 h. The reaction mixture was subsequently rotatory evaporated at an elevated temperature (*ca.* 90 °C) until the unspent reactants were removed from the product. The product (pale yellow oil) was used without further purification (see the Appendix A for the ^1^H and ^31^P spectra: Appendix A).
Yield = 45.5 g (95%)

^1^H NMR (600 MHz, CDCl_3_): ∂ 7.27 (m, 5H, Ar), 3.98 (m, 4H, -P-O-CH_2_-CH_3_), 3.14 (d, 2H, -P-CH_2_-Ar), 1.21 (t, 6H, -P-O-CH_2_-CH_3_)^31^P NMR (243 MHz, CDCl_3_): ∂ 26.41GC/MS: Retention time = 8.00 min; [M]^+.^− 137 = 91 (benzylic radical: corresponding to the most abundant species)

### 2.3. A Typical Procedure for Bulk Polymerization 

The procedure was adopted from a previously reported work with minor variations [3]. In this method, the required amount of monomer(s) and initiators were stirred thoroughly in a conical flask under a nitrogen atmosphere for ca. 1 h at 70–80 °C, until a visible increase in the viscosity was observed. Subsequently, the required amount of the additive/reactive was added and stirred for another 1 h, and the mixture was subsequently poured into an aluminium pan of ca. 50 mL volume and the pan stoppered with an aluminium lid. The pan was placed in an air oven preheated at 40 °C and kept for curing for about 20 h. During the second stage of curing, the temperature of the oven was raised to 60 °C for 8 h. Finally, after another 20 h of curing at 80 °C, the pan was cooled to room temperature. The final solid plaque, in the shape of the aluminium pan, was extracted from the pan. In the present study, a fixed phosphorus loading of 2 wt% was used in the case of each additive/reactive. 

### 2.4. Nuclear Magnetic Resonance (NMR) Spectroscopy

To obtain the purity and structure of the additives, precursors, monomers and polymeric materials, a Bruker 600 MHz instrument, Sydney, Australia, was employed, and the spectra were run in deuterated solvent (CDCl_3_ or DMSO-*d_6_*) at ambient probe conditions (see also the Appendix A). For ^31^P NMR the signals were calibrated against phosphoric acid as the external calibrant. The raw data were subsequently processed by using proprietary software from the manufacturer (TopSpin 4.0.8, Bruker Corporation, Sydney, Australia).

### 2.5. Thermogravimetric Analysis (TGA)

The TGA runs on the polymeric products were run, in a nitrogen atmosphere, at 10 °C min^−1^ and 60 °C min^−1^, from 30 to 900 °C, using a Mettler-Toledo instrument, Melbourne, Australia, with a gas flow rate of 50 mL min^−1^. The runs were done in triplicate, and it was found to be highly reproducible in that the associated thermograms were found to be perfectly overlapped on each other. The set heating rate of 60 °C min^−1^ was chosen to compare and correlate the results from the TGA experiments to those of other calorimetric techniques, such as pyrolysis combustion flow calorimetry (PCFC).

### 2.6. Differential Scanning Calorimetry (DSC) 

In the present study, DSC runs were primarily used to estimate the heat of pyrolysis for the various polymeric materials. For this purpose, the thermograms were recorded in a nitrogen atmosphere, at a heating rate of 10 °C min^−1^, from 30 to 550 °C, using a Mettler-Toledo instrument. The reproducibility of the DSC tests was optionally checked and found to be quite acceptable. The glass transition temperatures of the parent and modified polymers were also deduced from the DSC thermograms.

### 2.7. Pyrolysis Combustion Flow Calorimetry (PCFC) 

This technique, also known as ‘microscale combustion calorimetry’ (MCC), is a small-scale calorimetric testing method (milligram-scale) used to analyse the fire behaviour of various solid materials when subjected to forced non-flaming combustion, under anaerobic or aerobic conditions (ASTM D7309) [24,25,26]. The test method often provides information regarding useful combustion parameters of the test sample, such as peak heat release rate (pHHR), temperature to pHRR, total heat released (THR), heat release capacity (HRC), effective heat of combustion (*h_c_*) and percentage of char yield. In the present work, PCFC runs were carried out in some chosen substrates (mainly step-growth polymers) at 1 °C min^−1^, using an FTT microscale calorimeter using method A- i.e., in an atmosphere of nitrogen.

### 2.8. ‘Bomb’ Calorimetry

‘Bomb’ calorimetry measurements were performed on an IKA C200 instrument (IKA, Oxford, UK). Pelleted samples, weighing ca. 0.5 g, were placed inside a ‘bomb’ cell. The ‘bomb’ was filled with pure oxygen up to 30 bar, and the sample was subsequently ignited. The instrument was periodically calibrated using recrystallized benzoic acid. The final calorific values were displayed by the instrument using built-in software. For each sample, triplicate runs were performed.

### 2.9. Software Used for TGA 

This method was based on an algorithm and its accompanying software that was reported previously [27,28]. In this approach, one of the non-isothermal thermograms is chosen [29]. In the present study, in all cases, the thermograms obtained at a relatively low heating rate of 10 °C min^−1^ were chosen as this is expected to capture most of the underlying steps in the thermal degradative pathway of the substrate in question. As the first step, the data comprising the thermogram were transferred into an Excel file for subsequent processing, which primarily involved identifying the main step of decomposition.

## 3. Results and Discussion

The most important use of MMA-based polymers is evidently as transparent plaques for various applications, where good optical clarity and enhanced weather resistance are the main prerequisites. However, for such applications, the relatively high flammability of virgin PMMA often becomes a limiting factor. In this context, plaques of both PMMA were prepared (*ca*. 50 g scale) by incorporating various additives/reactives by adopting some previously reported procedures [3,30]. It is also relevant to note that the loading of phosphorus, in all cases, was normalized to 2 wt%, while altering the chemical environments and oxidation states of the phosphorus atom (III, or V) in the admixtures. Furthermore, both the *additive* and *reactive* routes were utilized, to identify the influences, if present, between the two strategies on the combustion features of the polymeric products. Furthermore, given the relatively nominal loading of phosphorus (2 wt%), any marginal improvements in the fire retardance of the modified systems as compared to the virgin polymers would be most advantageous. 

The products obtained through the bulk polymerization route were also chosen for further and detailed investigations in terms of their thermal (TGA) and calorimetric (DSC) properties, as well as their combustion (PCFC and ‘bomb’ calorimetry) characteristics. It is worth noting here that the final compositions of these products through the *reactive* strategy were effectively controlled with a high degree of certainty since the polymerization reactions were driven to near completion. Basically, this was achieved by using a mixture of low- and high-temperature initiators (benzoyl peroxide and dicumyl peroxide, respectively), and checking the structure of the final products through spectroscopic means (FT-IR and ^1^H NMR). The spectra so obtained were found to be devoid of any discernable signals from the residual monomeric species indicating that the polymerizations proceeded to ca. 99% conversion (see also in the Appendix A). As expected, the products exhibited relatively high purity, when compared to polymers made through the other common chain-growth techniques, since the bulk polymerization method does not require any solvent(s)/reagent(s).

In almost all cases, dense and tough polymeric plaques were formed. However, some exceptions can be noted. The DOPO-modified version of PMMA was found to be substantially brittle and amenable to shattering quite easily under a mechanical strain. On the other hand, the reactively modified version with the P- and N-containing co-monomer, ADEPMAE, was found to have a plasticizing effect on the final product. 

### 3.1. Thermogravimetric Analysis (TGA)

In this section, the relevant TGA parameters obtained for each PMMA-based sample at a heating rate of 10 °C min^−1^ are presented in Table 3. All the runs were carried out in a nitrogen atmosphere. The thermograms obtained for PMMA samples generally exhibited only one main decomposition step. However, during the initial phases of decomposition, some small mass losses can be also observed in almost all the cases. 

It can be noticed from the above table that the homopolymer (PMMA) has the highest temperature to initial mass loss (i.e., the induction point), followed by PMMA + TPPO. The corresponding value is least for PMMA + DEHPi (starting at 52 °C), which can be attributed to the release of some small molecules. However, this polymeric system produced the highest amount of char among the additives. On the other hand, PMMA + DEpVBP produced the maximum quantity of char among the reactives selected for the study. In addition, the reactively-modified systems generally exhibited relatively high thermal stabilities, in terms of the temperature at 50 wt% composition (almost 30 degrees higher compared to pure PMMA), than those systems containing the additives. This can be attributed to the interference of the acrylic co-monomeric unit with the unzipping reactions of sequential parts of the methacrylate units, thereby partially blocking the usual pathway of complete decomposition of the PMMA chains to yield the monomer [3]. Furthermore, thermal cracking of the pendent phosphorus-containing moieties has been shown to be responsible for the condensed-phase mechanism which results in char formation, through the initial formation of substituted phosphorus acids. Previous studies have already identified that reactive compounds are generally more effective than additives [3,30]. Generally, a higher char yield in a TGA run can be indicative of an enhanced degree of activity of the modifying group in question in the condensed phase. The overlays of the thermograms are provided in Figure 1, Figure 2, Figure 3, Figure 4, Figure 5 and Figure 6 (the corresponding data obtained at 60 °C min^−1^ were primarily used for comparison with the relevant parameters obtained through the PCFC runs).

### 3.2. Kinetic Analysis of the TGA Thermograms

As the first step, data of the unmodified PMMA bulk sample were initially used as the input for the software to establish the best-suited kinetic model [27,28,29]. From the run, it was found out that PMMA followed the two-dimensional diffusion (D2) model (a typical value for PMMA is shown to be 180 kJ mol^−1^). Therefore, for the ensuing analyses, this model was applied. The values of the apparent activation energy (*E_a_*), thus obtained, for the various PMMA-based bulk samples are given in the table below (Table 4).

It should be noted here that the apparent activation energy obtained in each case was an average value, over the entire region pertaining to the main step of the decomposition, as observed in the respective thermograms; hence, the spread of *α* values was also correspondingly chosen (see the entries in the last column of Table 4). In most cases, a reduction in the activation energies in the case of the modified samples can be observed. It is also relevant to note that the average values of *E_a_* can also be influenced by their relatively lower *α* values (where *α* is around 0.1 to 0.2). Here it is to be assumed that noticeable degrees of volatilization of low-molecular-weight additive (typically, in the case of some liquid compounds) can occur, where the energetic requirements are much lower than those required to break typical covalent bonds. Such an effect also seems to be reflected in the correspondingly lower values for the induction temperatures in the thermograms. 

Here, virgin PMMA is taken as a reference point in order to identify the extent of spread of the *E_a_* values. The DOPO modified PMMA is found to have the highest average value of the *E_a_* (235 kJ mol^−1^). All the other samples exhibited a lower apparent activation energy than unmodified PMMA. This seems to suggest that the presence of DOPO enhanced the thermal stability of PMMA, whereas in all other cases the modifications were found to decrease this.

### 3.3. Differential Scanning Calorimetry

Heat of pyrolysis (∆*H_pyro_*) data for various PMMA- based bulk samples were obtained from the DSC runs of the samples at a heating rate of 10 °C min^−1^ in the temperature range of 30–550 °C. The samples were accurately weighed into stoppered aluminum pans, and subsequently, a pinhole was made to release any excess pressure due to the emanating gaseous species, if present. The ∆*H_pyro_* values in each case were calculated by normalizing the area of the main pyrolysis peak (in mJ) with the mass of the sample (in mg). Given below are the results obtained for the PMMA-based bulk samples from the DSC runs as well as their corresponding ∆*H_pyro_* values (see Table 5).

It can be observed from Table 5 that PMMA modified with TPPO, TEPa, DEPP, DEBP (additives), and DEAEPa and DEpVBP (reactives), exhibited lower values of ∆*H_pyro_* than the unmodified material. The lowest value observed was that for PMMA + DEAEPa (274 mJ mg^−1^). On the other hand, the incorporation of the additives TPP, DOPO, DEHPi, TEPi and the copolymerization with the monomers DE-1-AEP and ADEPMAE has resulted in higher values of ∆*H_pyro_*, with PMMA + DEHPi being the highest (1030 mJ mg^−1^). The relatively widespread values for heat of pyrolysis (essentially endothermic in nature) among the various systems can be thought to arise from: (i) the difference in the energy requirements accompanying the phase changes of the additives (i.e., enthalpy of vapourization); (ii) the energetic needs for bond cleavage(s) of the polymeric chains; and (iii) thermal energy requirements for the cracking of the pendent modifying group and/or altered paths of the main chain decomposition process(s), as in the case of reactively modified systems. It can be clearly noted from the above table that the modifying groups (both in the case of additives and reactives) had an effect on the glass transition temperature of the parent polymer matrix. It can be further noted here that in all cases these groups seem to exert a plasticizing effect, thus the corresponding Tg values are lower as compared to the base polymer.

### 3.4. Pyrolysis Combustion Flow Calorimetry (PCFC)

The PCFC runs for each bulk sample were performed as triplicates in order to ensure their reproducibility, and the average values are presented below in Table 6.

It can be noticed from the table that the values of pHRR, HRC, THR and EHC of the modified samples vary to different extents, with some modified versions having higher values than the unmodified PMMA, whereas others having lower values. For example, in the case of solid additives, a noticeable reduction in the value of pHRR was observed for the TPPO modified version (277 W g^−1^), whereas the material incorporating DOPO showed a higher value (413 W g^−1^). The systems with the liquid additives, i.e., PMMA with DEHPi and TEPi, also showed relatively high pHRR values (493 and 439 W g^−1^, respectively). This can be attributed to the higher volatility of the liquid samples compared to the solid versions, and that they can contribute to the fuel load, rather than showing combustion inhibitory effects, especially, in the vapor phase. The other liquid additives used in the present study, including TEPa, DEPP and DEBP, seem to assist in reducing the pHRR values of the polymer matrix (276, 253 and 294 W g^−1^, respectively). The difference in the combustion inhibitory effects amongst the liquid additives can be attributed to their: volatility; chemical natures; decomposition pathways, and hence the nature and composition of the volatiles; etc. Amongst the reactively modified versions, a slight decrease in the pHRR value is only observed in the case of the sample with DEAEPa (the phosphate monomer) which tends to form phosphoric acid species more easily [18,30] and a noticeable drop in the case of DEpVBP, a styrenic-type monomer that can enhance the formation of condensed aromatic species [12]. 

The evolution of volatile species, during the early stages of the TGA thermograms, is also reflected in the HRR curves as shoulder peaks, indicating that these fragments in effect act as fuel loads for combustion reactions in the latter stage of the PCFC tests (i.e., the second stage that involves aerobic and forced combustion at ca. 750 °C). This attribute was also clearly evident in the first derivative of the TGA thermograms, which favorably compared to the HRR curves, in their general and overall profiles. In short, under a programmed heating regime (i.e., 1 °C s^−1^ in nitrogen, which is similar to a heating rate of 60 °C min^−1^ in TGA) as encountered in the first phase of the PCFC test, the modifying groups (both the additive compounds or the reactive versions, as the case may be) do not seem to interact co-operatively with the decomposition of the underlying parent polymer matrix, in that they seem to fail to induce any combustion inhibitory effects. 

### 3.5. ‘Bomb’ Calorimetry

The values of the heat of combustion, Δ*H_comb_*, obtained through the ‘bomb’ calorimetric runs are collected in Table 7.

The values of the heat of combustion for the modified systems, as compared to the parent polymer, are expected to be lower when combustion inhibitory effects are in operation. However, such an effect was only noticeable in the case of PMMA with the additives, TPP, DEHPi, TEPi, TEPa, and with the reactives DE-1-AEP and ADEPMAE. Among these, the PMMA + DEHPi sample showed the lowest value (24.87 kJ g^−1^). Hence, it is to be assumed that some additive/reactive groups, upon decomposition, produce volatiles which, in turn, can exert some degree of combustion inhibition. However, in the case of other modified samples (with TPPO, DOPO, DEPP and DEBP, and with the reactive monomer DEpVBP), slightly higher values of the heat of combustion were observed. Therefore, it can be inferred here that these additive/reactive groups, during decomposition, result in the formation of volatile species that are combustible in nature, thus increasing values for the total heat of combustion in such systems, relative to that of the parent polymer. 

### 3.6. Some Generalizations among the Test Parameters

Some generalizations were found to exist amongst some of the relevant test parameters from TGA, PCFC and ‘bomb’ calorimetry techniques. It can be noted that the TGA runs yielded less than 1 wt% in all cases, whereas the corresponding values in the case of PCFC tests varied widely (from 0 to 6 wt%, except in the case of the P/N monomer: 11 wt%).

The EHC values calculated from PCFC data, where only incomplete and forced non-flaming combustion is affected, are lower than the corresponding ∆*H_comb_* values obtained through ‘bomb’ calorimetric runs, where the ‘complete’ combustion of the sample is assumed to occur. Furthermore, the degree deviances between each of the samples also varied widely. This can be attributed to the fact that the chemical nature of the modifying compounds/groups and the possible modes of interaction of the modifying agents with the base substrate are also different in each case.

## 4. Conclusions

The synthetic strategies employed in the current work proved to be quite facile and resulted in the desired products with an acceptable level of purity in each case. Furthermore, the bulk polymerization route that was adopted in the present work produced solid plaques, and there was enough spectroscopic evidence to suggest near-complete polymerization of the monomer(s). From the results obtained through the investigations, the following inferences can be drawn in the case of PMMA-based bulk systems: (1) TGA: Generally, the products obtained through the reactive strategy exhibited improved thermal stabilities and, in particular, the copolymer containing the P/N-monomer, ADEPMAE, was found to be the most effective. On the other hand, such effects greatly varied among the modified polymers containing the additives; (2) DSC: The values of the heats of pyrolysis obtained from the DSC measurements were found to be spread out, where some systems showed higher values whilst the rest exhibited lower values when compared to virgin PMMA. Here it should be noted that the modifying groups (both the additive and reactive ones) exerted a noticeable plasticizing effect on the parent polymer matrix as gauged by the reductions in the values for the glass temperature; (3) PCFC: The relevant parameters, such as pHRR, HRC, THR and EHC of the modified samples also varied to different extents, where some modified versions showed higher values than the unmodified PMMA, whereas others exhibited lower values. This essentially points out the differences in the combustion inhibitory effects, if present, of the additives/reactive components employed for the study. This can be attributed to the specific chemical environment/oxidation state of the phosphorus atom in each case; (4) ‘Bomb’ calorimetry: The heat of combustion values obtained through this method also varied among the test samples, some of which showed higher values compared to PMMA, whereas the corresponding values were lower in the case of others. This essentially points to the fact that the different modifying groups exerted varying nature and degree of combustion behaviors, in some cases an additive, and even antagonistic, effect. It is relevant to note that such effects were also observed in the relevant parameters obtained through the PCFC measurements.

The utility of in-house developed software, in deducing the Arrhenius parameters, from the TGA runs was also explored and successfully applied. However, given that such values were deduced by considering only a thermogram obtained through a single heating rate, the extended validity of these values should be treated with caution, and such values can only be treated as ‘apparent’ values at best. Obviously, a more thermodynamically robust approach needs to incorporate multiple heating rates and/or isothermal runs at the chosen temperature values. The rationale behind such an approach, as compared to the corresponding results obtained through some of the conventional methods that are based on multiple heating regimes, is discussed in detail in a previous publication [29]. However, the apparent values obtained in the present investigation, through a single heating rate, can be applied for comparative purposes amongst the various modified systems as against the virgin material with a degree of caution. The findings relating to the mode of action of the P-containing additives/reactive, in the gaseous- and condensed-phase, operating in these systems are published elsewhere [31,32].

## Figures and Tables

**Figure 1 polymers-14-01447-f001:**
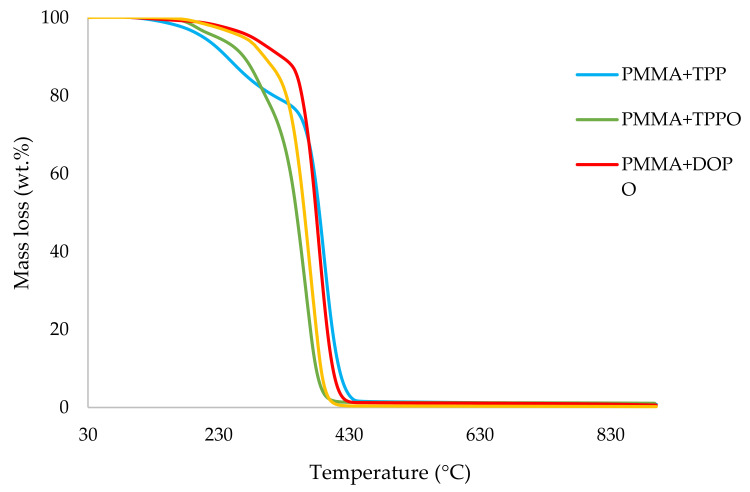
An overlay of the TGA curves of the PMMA-based materials with solid additives, at 10 °C min^−^^1^.

**Figure 2 polymers-14-01447-f002:**
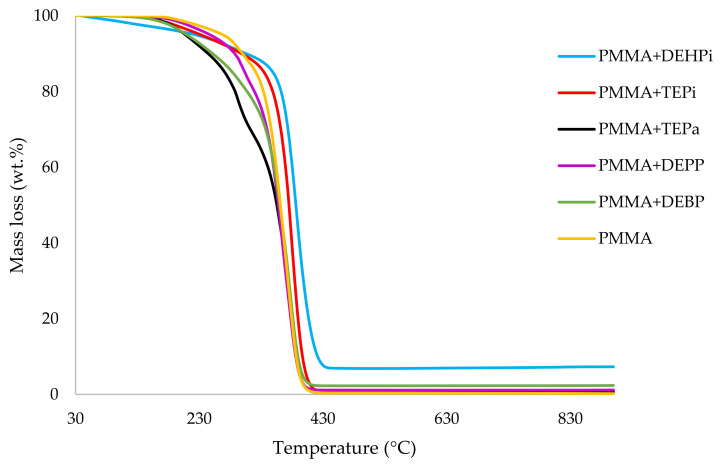
An overlay of the TGA curves of the PMMA-based materials with liquid additives, at 10 °C min^−1^.

**Figure 3 polymers-14-01447-f003:**
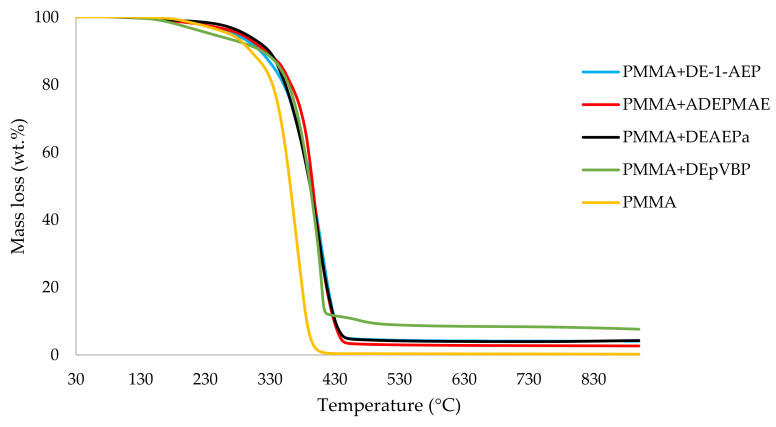
An overlay of the TGA curves of PMMA-based materials with reactives, at 10 °C min^−1^.

**Figure 4 polymers-14-01447-f004:**
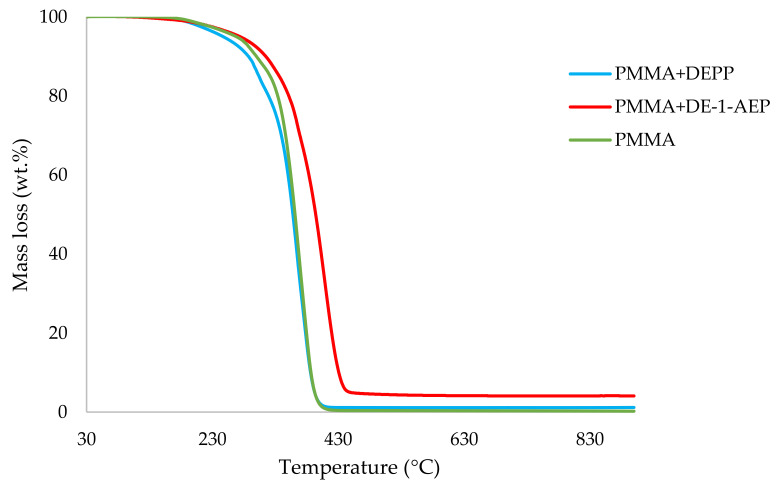
An overlay of the TGA curves of PMMA and PMMA + aliphatic phosphonate materials.

**Figure 5 polymers-14-01447-f005:**
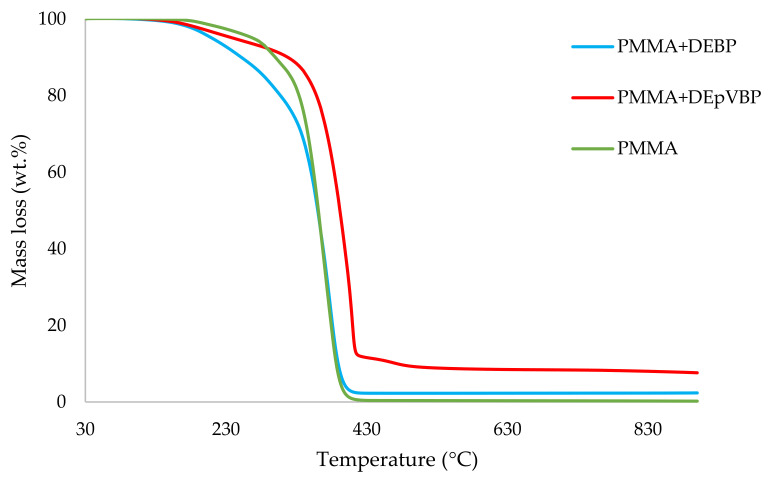
An overlay of the TGA curves of PMMA and PMMA + aromatic phosphonate materials.

**Figure 6 polymers-14-01447-f006:**
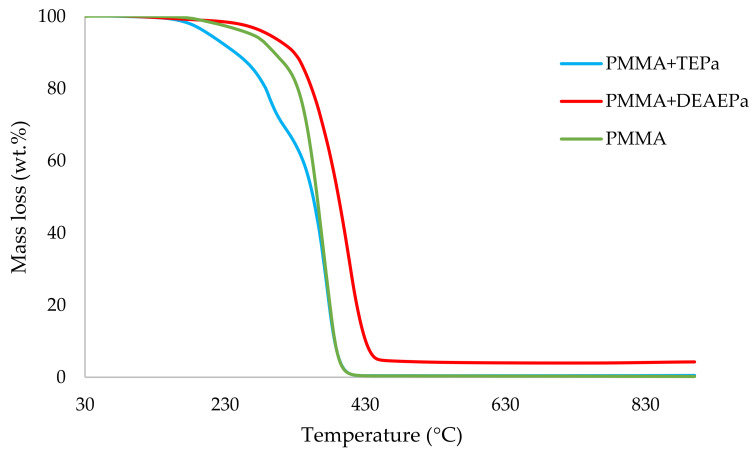
An overlay of the TGA curves of PMMA and PMMA + phosphate materials.

**Table 1 polymers-14-01447-t001:** The additives and reactives used for the bulk polymerization of MMA.

Sl. No.	Additive/Reactive	Structure/Oxidation State
1.	Triphenylphosphine (TPP), *additive*	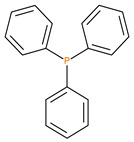
2.	Triphenylphosphineoxide (TPPO), *additive*	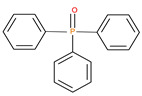
3.	9,10-Dihydro-9-oxa-10-phosphaphenenthrene-10-oxide (DOPO), *additive*	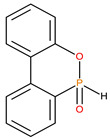
4.	Diethylphosphite (DEHPi), *additive*	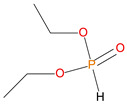
5.	Triethylphosphite (TEPi), *additive*	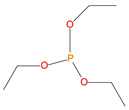
6.	Triethylphosphate (TEPa), *additive*	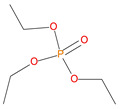
7.	Diethylpropylphosphonate DEPP), *additive*	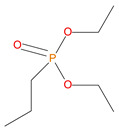
8.	Benzylphosphonate (DEBP), *additive*	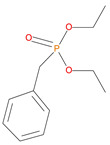
9.	Acrylic phosphonate (DE-1-AEP), *reactive*	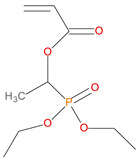
10.	P/N (ADEPMAE), *reactive*	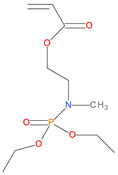
11.	Acrylic phosphate (DEAEPa), *reactive*	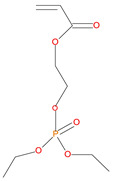
12.	Vinylbenzylphosphonate (DEpVBP), *reactive*	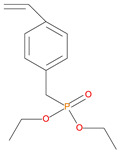

**Table 2 polymers-14-01447-t002:** Details of the preparative data for the bulk polymers.

Sl. No.	MMA (ml)	Additive/ Reactive	Formula Weight	Additive/Reactive Weight (g/mL)	BPO/Dicumyl Peroxide (mg)
1	40.00	---	---	--	40.0/20.0
2	44.20	TPP	262	8.46 g	54.0/27.0
3	43.65	TPPO	278	8.97 g	53.0/26.5
4	45.80	DOPO	216	6.97 g	53.0/26.5
5	48.46	Diethylphosphite	138	4.15 ml	52.0/26.0
6	47.50	Triethylphosphite	166	5.50 ml	53.0/26.5
7	46.95	Triethylphosphate	182	5.48 ml	53.0/26.5
8	37.61	Diethylpropylphosphonate	180	4.65 g	43.0/21.5
9	36.29	Diethylbenzylphosphonate	228	5.88 g	44.0/22.0
10	22.50	DE-1-AEP monomer	236	3.81 g	26.0/13.0
11	22.05	ADEPMAE monomer	265	4.27 g	26.0/13.0
12	22.30	DEAEPa monomer	252	4.06 g	26.0/13.0
13	22.24	DEpVBP monomer	254	4.10 g	26.0/13.0

**Table 3 polymers-14-01447-t003:** Some relevant parameters from the TGA analyses of PMMA-based systems.

Sl. No.	Sample	Induction Temp (°C)	Temp at 50 wt% (°C)	Residue at 500 °C (wt%)	Final Residue at 800 °C (wt%)
1	PMMA	157	362	0.4	0.3
2	PMMA + TPP	93.0	385	1.4	1.0
3	PMMA + TPPO	147	352	1.3	1.2
4	PMMA + DOPO	88.0	380	1.2	0.9
5	PMMA + DEHPi	52.0	387	6.9	7.2
6	PMMA + TEPi	74.0	375	0.7	0.7
7	PMMA + TEPa	83.0	356	0.4	0.5
8	PMMA + DEPP	98.0	358	1.1	1.1
9	PMMA + DEBP	89.0	359	2.3	2.3
10	PMMA + DE-1-AEP	97.0	395	4.5	4.1
11	PMMA + ADEPMAE	98.0	396	3.1	2.7
12	PMMA + DEAEPa	103	393	4.3	4.0
13	PMMA + DEpVBP	107	392	9.2	8.1

**Table 4 polymers-14-01447-t004:** Values of the apparent energy of activation (*E_a_*), Arrhenius parameter (*A*), and other relevant parameters of PMMA-based samples obtained using the software (model: D2 Two-dimensional diffusion).

Sl. No.	Sample	Apparent Activation Energy (*E_a_*, kJ mol^−1^)	A (min^−1^)	*R^2^* Values	*α*-Value Range
1	PMMA	183	1.11 × 10^14^	0.9399	0.1 to 0.9
2	PMMA + TPP	139	3.34 × 10^10^	0.9991	0.3 to 0.9
3	PMMA + TPPO	114	2.19 × 10^8^	0.9277	0.1 to 0.9
4	PMMA + DOPO	235	6.61 × 10^17^	0.9871	0.1 to 0.9
5	PMMA + DEHPi	125	4.00 × 10^8^	0.9038	0.1 to 0.9
6	PMMA + TEPi	173	3.46 × 10^13^	0.9963	0.2 to 0.9
7	PMMA + TEPa	101	6.15 × 10^7^	0.9714	0.4 to 0.9
8	PMMA + DEPP	137	1.61 × 10^10^	0.9344	0.1 to 0.9
9	PMMA + DEBP	109	2.67 × 10^8^	0.9796	0.3 to 0.9
10	PMMA + DE-1-AEP	140	6.76 × 10^9^	0.9526	0.1 to 0.9
11	PMMA + ADEPMAE	169	1.30 × 10^12^	0.9475	0.1 to 0.9
12	PMMA + DEAEPa	160	2.96 × 10^11^	0.9514	0.1 to 0.9
13	PMMA + DEpVBP	121	9.13 × 10^8^	0.9878	0.2 to 0.8

**Table 5 polymers-14-01447-t005:** Heat of pyrolysis data of PMMA-based materials obtained from DSC tests.

Sl. No.	Samples	Heat of Pyrolysis, ∆*H_pyro_* (mJ mg^−1^)	Tg (°C) (±5 °C)
1	PMMA	420.0	120
2	PMMA + TPP	660.0	70
3	PMMA + TPPO	330.0	110
4	PMMA + DOPO	640.0	70
5	PMMA + DEHPi	1030	70
6	PMMA + TEPi	790.0	80
7	PMMA + TEPa	320.0	90
8	PMMA + DEPP	300.0	70
9	PMMA + DEBP	310.0	83
10	PMMA + DE-1-AEP	450.0	90
11	PMMA + ADEPMAE	680.0	65
12	PMMA + DEAEPa	274.0	99
13	PMMA + DEpVBP	340.0	79

**Table 6 polymers-14-01447-t006:** PCFC data of PMMA-based materials.

Sl. No.	Samples	Temp to pHRR (°C)	pHRR (W g^−1^)	THR (kJ g^−1^)	HRC (J g^−1^ K^−1^)	Char Yield (wt%)	EHC (kJ g^−1^)
1	PMMA	386	358	22.8	358	0.0	22.8
2	PMMA + TPP	405	339	23.2	343	4.3	24.2
3	PMMA + TPPO	373	277	23.8	283	2.9	24.5
4	PMMA + DOPO	398	413	24.8	414	0.4	24.9
5	PMMA + DEHPi	393	493	22.2	493	3.2	22.9
6	PMMA + TEPi	397	439	21.4	439	1.4	21.7
7	PMMA + TEPa	388	276	20.6	277	0.4	20.7
8	PMMA + DEPP	387	253	22.0	254	0.8	22.2
9	PMMA + DEBP	386	294	20.4	303	1.4	20.7
10	PMMA + DE-1-AEP	399	367	22.4	368	2.7	23.0
11	PMMA + ADEPMAE	420	373	22.1	372	11	24.9
12	PMMA + DEAEPa	426	336	22.3	343	0.0	22.3
13	PMMA + DEpVBP	432	271	22.5	308	5.4	23.8

**Table 7 polymers-14-01447-t007:** Heat of combustion data for PMMA-based samples from ‘bomb’ calorimetric measurements.

Sl. No.	Sample *	Δ*H_comb_* (kJ g^−1^)
1	PMMA	26.24
2	PMMA + TPP	26.12
3	PMMA + TPPO	27.12
4	PMMA + DOPO	26.41
5	PMMA + DEHPi	24.87
6	PMMA + TEPi	25.42
7	PMMA + TEPa	25.74
8	PMMA + DEPP	26.55
9	PMMA + DEBP	26.56
10	PMMA + DE-1-AEP	25.27
11	PMMA + ADEPMAE	25.80
12	PMMA + DEpVBP	26.49

* The ∆*H_comb_* of PMMA+ DEAEPa could not be performed.

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
