# Peer review of "Thermal and Calorimetric Investigations of Some Phosphorus-Modified Chain Growth Polymers 1: Polymethyl Methacrylate"

_polymers, 2022, doi:10.3390/polym14071447_

Round 1

Reviewer 1 Report

In the present paper, the thermal and calorimetric characterizations of chain-growth polymers, based on polymethyl methacrylate (PMMA) are reported. The results showed that the different modifying groups exerted varying nature and mode of combustion behaviors. However, I suggest that some major revision should be made. 1. there are some formatting and syntax errors. For example, Abstract: "even an antagonistic , effect". “2. 2.2. . Synthesis of the additive (diethylbenzylphosphonate)”. 2. How about the mechanical properties of the polymers? 3. "3. Results and discussion". Polymer photograph and their SEM images should be added to show the surface apparent morphology. 4. EDX for polymers and their residual carbon should be added, and flame-retardant mechanism should be elaborated. 5. Phosphorus-modified chain growth on the thermal and calorimetric of polymers should be mainly summarized and explained. 6. Please indicate the meaning of A in Table 4. 7. "4. Conclusions" should be rewritten.

Reviewer 2 Report

Authors described the thermal and calorimetric investigations of some phosphorus-modified chain growth polymers: polymethyl methacrylate. I consider that manuscript meets all requirements to be published after minor revision. The IR, 1H, and 13C NMR spectra of synthesized additives, monomers, and polymeric materials might be included in the Supplementary Material. A short explanation of relevant IR and NMR data might be included in the manuscript to confirm the preparation of the polymer. For instance, two Figures might be include in the manuscript to show IR and 1H/13C NMR spectra under difference reaction conditions to see the best result. Additional suggestions and comments are included:    

(1) See abstract. It is very important to complement this information with relevant experimental data.  

(2) See line 106. (0.19 mol; 95 mol%- i.e. near quantitative yield). I consider that 0.19 mol and 95 mol% might be substituted by 95%.

(3) See 2.2. Synthesis of the additive (diethylbenzylphosphonate). (a) Colour of the solid or oil might be included, (b) if it is a solid, the melting point might be included, (c) the 1H NMR reporting data might include the coupling constants JHP and JHP, (d) 13C NMR reporting data might be included, and (e) IR reporting data might be included.

(4) See line 132. (CDCl3 or DMSO-d6 instead of (CDCl3, or d6-DMSO).

(5) See lines 198 and 199. Authors mention that “checking the structure of the final products through spectroscopic means (FT-IR and 1H NMR). The spectra so obtained were found to be devoid of any discernable signals from the residual monomeric species indicating that the polymerizations proceeded to ca. 99% conversion”. However, a short explanation of most relevant IR and 1H and 13C NMR data might be included to correlacionate the successful of polymerization process.  

(6) See references. It is very important to check the guideline of “Polymers”.

(7) See Supplementary Material. This section is absent. The IR, 1H, and 13C NMR spectra of synthesized additives, monomers, and polymeric materials might be included. A short explanation of relevant data might be included in the manuscript to confirm the synthesis of the expected polymer.  

(8) IR and 1H and 13C NMR reporting data of the synthesized polymer might be included in the 2. Materials and Methods.  

Author Response

Please see attached word document

Round 2

Reviewer 1 Report

I am pleased to accept this version.